# Motivators, Barriers, and Acceptance of COVID-19 Vaccination among Residents of Western Saudi Arabia

**DOI:** 10.3390/vaccines10122097

**Published:** 2022-12-08

**Authors:** Abdulaziz A. Alhothali, Waleed F. Alotaibi, Bassam L. Alabbadi, Yousef F. Alsubaie, Ahmed Ibrahim Fathelrahman, Asim Ahmed Elnour, Azza A. K. El-Sheikh, Sayed F. Abdelwahab

**Affiliations:** 1College of Pharmacy, Taif University, Taif 21944, Saudi Arabia; 2Department of Clinical Pharmacy, College of Pharmacy, Taif University, Taif 21944, Saudi Arabia; 3College of Pharmacy, Al Ain University, Abu Dhabi Campus, Abu Dhabi, United Arab Emirates. AAU Health and Biomedical Research Center, Al Ain University, Abu Dhabi P.O. Box 112612, United Arab Emirates; 4Basic Health Sciences Department, College of Medicine, Princess Nourah bint Abdulrahman University, Riyadh 11671, Saudi Arabia; 5Department of Pharmaceutics and Industrial Pharmacy, College of Pharmacy, Taif University, Taif 21944, Saudi Arabia

**Keywords:** COVID-19, COVID-19 pandemic, COVID-19 vaccine barriers, SARS-CoV-2, side effects, vaccines, vaccine acceptance

## Abstract

Background: There are limited studies that have assessed COVID-19 vaccine acceptance and side effects, both globally and in the western region of Saudi Arabia (SA). Objective: This study assessed the acceptance of vaccination against COVID-19, determined motivators and barriers for taking these vaccines, and assessed vaccine side effects in the western region of SA. Study design: The study was an online cross-sectional study conducted among the people who lived in the western region of SA during the period from December 2021 to March 2022. Participation was voluntary for participants who were above 18 and lived in the Western region of SA. Children and those living in other countries were excluded from the study. Methods: The study tool was a self-administered questionnaire which assessed COVID-19 vaccine acceptance, determined motivators and barriers for taking the vaccines, and assessed their side effects among 1136 participants in the western region of SA. Data gathered were analyzed by the SSPS version 22 software. Result: A total of 1136 individuals, aged 18 years and above, participated in the study, with 50.7% (*n* = 567) being males. Most of the participants were from Taif city (68.4%; *n* = 777), and 57.6% (*n* = 654) were unmarried. Pfizer was the most frequently administered vaccine (72.8%; *n* = 823). Most participants explained that their vaccine administration protected themselves and their families (70.5%; *n* = 835). The acceptance showed that 55% (*n* = 626) of the participants had either very high or high confidence in the efficacy of the COVID-19 vaccines, while 14.7% (*n* = 167) of them had low/very low confidence in its efficacy. The side effects showed that 80.8% (*n* = 918) of the participants showed that they did not have any difficulties attributed to COVID-19 vaccine administration. Positive attitudes and practices were apparent, and most of the participants (78.3%; *n* = 889) tended to be actors in the fight against COVID-19. Conclusions: The current study showed a high level of acceptance of COVID-19 vaccination among people living in the western region of SA. Health education and communication from authoritative sources will be important to alleviate public concerns about COVID-19 vaccine safety.

## 1. Introduction

The first case of COVID-19 was reported in Wuhan in China in December 2019. Until now, there have been 623,121,528 confirmed cases of COVID-19, involving 6,549,730 deaths, as reported by the World Health Organization [WHO]. In Saudi Arabia [SA], the first confirmed case of COVID-19 was announced on 2 March 2020. Up to 1 October 2022, there have been 801,600 confirmed cases of COVID-19 infections and 9352 deaths in SA reported to the WHO [1,2]. The 2019 coronavirus disease [COVID-19] pandemic, caused by severe acute respiratory syndrome Coronavirus-2 [SARS-CoV-2], is a major threat worldwide. COVID-19 cases were characterized by the development of severe disease with a high mortality rate, especially among the elderly and those with comorbidities, such as cardiovascular diseases, chronic kidney disease, and chronic obstructive pulmonary disease [3].

The governments around the world have implemented various strict control measures for the COVID-19 pandemic. The authorities in SA adhered to strict measures, e.g., face masks, social distancing, partial and comprehensive closures, and the closure of schools and all business sectors. Although the impact is negative on the economic level, such measures have helped to flatten the epidemic curve. Nevertheless, the re-emergence and spread of COVID-19 (as well as the Delta and Omicron variants) have been reported. Therefore, there is an urgent need for long-term preventive measures. Few countries have sought to achieve herd immunity, which is defined as a level of immunity in a population that prevents outbreaks of disease through natural infection; however, such an approach has been deemed unethical [4]. Since the massive spread of COVID-19 and its impact, the world has faced difficulty in controlling the pandemic.

COVID-19 is a global threat due to its devastating effects on the world economy and healthcare systems. In addition, there is no approved treatment for this dangerous, highly infectious disease. This mandates the application of strict preventive measures and preventive vaccination campaigns. This is of specific importance within Middle Eastern countries during these difficult times of crises and political conflicts, where individuals usually fear financial difficulties, infection, isolation, lockdown, and death. This creates a state of psychological, behavioral, and physical distress among the population [5]. Currently, there are several available and emergency-approved vaccines against COVID-19 that increase immunity, which could help to end the pandemic. According to the WHO, the number of doses given around the world has reached approximately nine billion doses, and approximately four billion persons have been vaccinated with at least one dose. There are four vaccines approved in SA. These are Pfizer/BioNTech, Oxford/AstraZeneca, Moderna, and Janssen [Johnson and Johnson]. According to the WHO, the number of doses given in SA has reached 56 million doses, and approximately 25 million persons have been vaccinated with at least one dose [6,7]. The western region of SA has a population of approximately seven million people, representing almost one-fifth of the Saudi population. Based on the overall response, more studies should be conducted to evaluate the effectiveness and safety of vaccines. Many people expressed doubt as to whether substantial evidence was available for their safety, and the government created compulsion for vaccination by making it necessary for their jobs or travel welfare [8]. In this regard, effective vaccination campaigns are vital for successful control of the COVID-19 pandemic. Many people are concerned about the safety and efficacy of the new vaccine platforms. Therefore, measuring the population’s intent to get vaccinated is crucial. In this regard, there are limited studies that have assessed COVID-19 vaccine acceptance, motivators, barriers, and side effects in the Middle East, Gulf countries, and Saudi Arabia [9,10]. To our knowledge studies reporting data from Saudi Arabia suffer the limitation of having small sample sizes; thus, larger studies are needed. The present study was conducted to provide evidence in this area using a larger sample size. The study aimed to assess the motivators, barriers, and acceptance of COVID-19 vaccination among the residents of the western region of Saudi Arabia. The study showed a high level of acceptance of COVID-19 vaccination among the study participants.

## 2. Methods

### 2.1. Study Settings and Participants

The current study was a cross-sectional survey conducted among people living in the western region of SA who were conveniently invited to participate in this study. The study was conducted during the period from December 2021 to March 2022. Adults above the age of 18 who agreed to take part in the study were included. Participants under the age of 18 were not permitted. Participants were sent a link for the study survey, and participation was voluntary. The link was distributed to candidate participants on social media, including WhatsApp, Twitter, Telegram, and Facebook. Once the participant clicked on the study link, they were informed about the study on the first page. They were informed that their participation in the study was voluntary and they could exit the survey, if they needed so. Participants who were above 18 and lived in the western region of SA were included in the study, while children and those living in other regions were excluded from the study. The study protocol was approved by the ethics committee at Taif University, with approval number 43-312.

### 2.2. Sample Size Calculation

Based on the available statistics, we expected that about 5,000,000 people in the western region of SA received at least one dose of the COVID-19 vaccine [6,7]. As rule of thumb, a sample of 10 percent, recruited randomly, represents the population from which it is drawn, up to a sample size of 1000 people. Beyond that, there would be no need for a further increase (1000 would be representative of any size of population). We decided to recruit a slightly higher number (10–15%) than the 1000, as we expected that not all candidates who received the first dose of the COVID-19 received the second and third doses.

### 2.3. Measurement and Data Collection Tool

The current study tool was a self-administered questionnaire designed after consulting previously published studies, some of which having validated questionnaires [11,12,13,14]. Five sections made up the study questionnaire, all of which were created especially for it. Given that Arabic is the participants’ predominant language in Saudi Arabia, we gave the questions in Arabic for the best possible understanding. The final revised questionnaire contained five sections and 47 questions. Section one (ten items) was concerned with the demographic data, and participants were asked to indicate their gender, age, nationality, education level, residency city/area, marital status, and employment status. Section two contained five questions assessing the information about COVID-19 and its vaccines, including the participant’s administration of COVID-19 vaccine, vaccine type administered in the first and second doses of the vaccine, why the participant got the vaccine, and whether (s)he or a relative of hers/his had been infected or had died from COVID-19 infection. Section three assessed the vaccine acceptance and contained ten questions. Four questions were about the confidence in the vaccine, using a five-point Likert scale, and the remaining were yes/no/I do not know questions about vaccine acceptance. Section four assessed the motivators and barriers for COVID-19 vaccine administration, and comprised seventeen questions. These were yes/no/not applicable questions. Finally, section five contained four questions assessing the side effects of COVID-19 vaccines. The mean scores were generated for the questions from section three, except one question (rate of knowledge about vaccination) to identify the participants’ acceptance of COVID-19 vaccines.

The total acceptance rate was determined by taking the mean average acceptance response to the questions of Section 3, except for one question (rate of knowledge about vaccination).

### 2.4. Questionnaire Validity and Reliability

The questionnaire was examined by a group of four researchers from Taif University’s College of Pharmacy for the face and contents validity. They were asked to assess clarity, consistency, and suitability for the regional settings. Their suggestions were incorporated into the final version of the questionnaire. Additionally, 25 volunteers were used in a field test as a pilot sample to validate the questionnaire, and their data were excluded from the study.

Although all variables were intended to be analyzed individually, and there was no intention to compute a scoring instrument (i.e., a new variable composed of a group of items), we checked the reliability of the items assessing residents’ acceptance of the COVID-19 vaccination using the pilot sample data, and obtained a Cronbach’s Alpha of 0.873. For general interest and for the purpose of comparison, we tested the reliability of the items representing the motivators and barriers, and obtained Cronbach’s Alpha values of 0.341 and 0.466, respectively. Practically, it is not expected that such items would be internally consistent, because there would be variability between respondents in the variables representing the motivators and barriers.

### 2.5. Statistical Analysis

Statistical analysis was performed by using the Statistical Package for Social Sciences (SPSS) version 22 (IBM Corporation, New York, NY, USA). Descriptive statistics were generated for the responses and correlation coefficients to describe relationships between continuous variables. For independent variables, the Chi square test was used to compare categorical variables. A *p*-value of <0.05 was considered significant.

## 3. Results

### 3.1. Demographic Characteristics of the Study Participants

A total of 1136 participants successfully filled out the online questionnaire, and the responses were saved on a Google drive in a password protected manner. The baseline demographic characteristics of the participants are shown in Table 1. As shown, more than half of the participants (*n* = 669; 58.9%) were in the age range of 18–30 years (mean age: 34.5 SD: 9.8). Additionally, less than half of the participants were married (*n* = 458; 40.3%), and most of them had a college degree (*n* = 864; 76.1%). In this regard, male and female participants were equally distributed (Table 1). Due to the small numbers of participants aged above 60 years (*n* = 10), they were merged with the group aged 51–60 years. Both groups were included in a single group (>50 years), and were analyzed as such. Participants who did not work at the time of the study represented 62.0% (*n* = 704), and those who worked in the private sector comprised 9.9% (*n* = 112). Participants who had a family member working in the health sector represented around 40.1% (*n* = 455), and those who had seen or heard news and information about COVID-19 vaccination from social media (e.g., Facebook, Twitter, YouTube) made up 47% (*n* = 1047), in comparison to the 30% (*n* = 670) who heard about the pandemic from local television. Other details of the demographic data of the participants are shown in Table 1.

### 3.2. Participants’ Information about COVID-19 and Its Vaccine

Most of the participants had received the vaccine (*n* = 1054; 92.8%). The Pfizer vaccine was the most frequent vaccine taken by the participants (*n* = 827; 72.8%), followed by AstraZeneca (*n* = 256; 22.5%). Most people declared that the reason behind taking the vaccine was to protect themselves and their families (*n* = 835; 73.5%), followed by obtaining services that were restricted to those who had received the vaccine (*n* = 220; 19.3%). Interestingly, most of the participants (74.6%; *n* = 847) stated that they personally knew someone who had had COVID-19 infection or died from COVID-19 infection (Table 2).

### 3.3. Participants’ Motivators and Barriers for Administration of COVID-19 Vaccine

Table 3 shows the participants’ motivators and barriers. In total, 78.3% (*n* = 889) of participants wanted to be actors in the fight against COVID-19, and this was associated with residency area and marital status (*p* = 0.007 each). In addition, 63.4% (*n* = 720) of the participants wanted their children to resume school as soon as possible; this was associated with age (*p* < 0.001), residency (*p* = 0.004), marital status (*p* < 0.001), and employment (*p* < 0.001). In addition, 72.2% (*n* = 820) of the participants stated that they took the vaccine to avoid infection with COVID-19. Moreover, 95.9% (*n* = 1089) of the participants did not want to transmit COVID-19 infection to others. Furthermore, 39.8% (*n* = 452) of the participants stated that they took the vaccine because it was free, and this was associated with gender (*p* = 0.005), age (*p* < 0.001), educational level (*p* = 0.019), marital status (*p* < 0.001), and employment (*p* = 0.002).

Regarding mandatory vaccination, half of the participants (50.7%; *n* = 576) did not take vaccine because it was mandatory for traveling abroad, which correlated with gender (*p* = 0.019), educational level (*p* = 0.037), and employment (*p* = 0.003). In addition, one third of the participants (33.3%; *n* = 378) stated that their doctor’s recommendation was an important factor in vaccination decision-making, and this correlated with age (*p* = 0.008) and education level (*p* = 0.004). In addition, 58.9% (*n* = 669) of the participants stated that they were afraid of adverse effects of the vaccine, which correlated with gender (*p* < 0.001). Moreover, 59.4% (*n* = 675) of the participants indicated that vaccine convenience (vaccination method, frequency, distance to vaccination site, etc.) was an important factor in vaccination decision-making, which correlated with residency (*p* = 0.020) and marital status (*p* = 0.002). Approximately one half of the participants (51.6%; *n* = 586) showed that they did not really understand how the COVID-19 vaccine worked, and this was associated with age (*p* = 0.011), education level (*p* = 0.003), marital status (*p* = 0.019), and presence of a family member working in the health sector (*p* = 0.018). Moreover, 41% (*n* = 466) of the participants did not think that the development of the COVID-19 vaccines was too fast, which can be a barrier for vaccination, and this correlated with marital status (*p* = 0.009). 

### 3.4. Participants’ Acceptance of COVID-19 and Its Vaccine

The rate of acceptance for our participants to vaccination was 53.2%. As shown, 55% (*n* = 626) of the participants had very high or high confidence in the efficacy of the COVID-19 vaccines, while 14.7% (*n* = 167) of them had low/very low confidence in its efficacy. Confidence in the efficacy of the COVID-19 vaccines was associated with the gender of the participants (*p* < 0.001), their educational level (*p* = 0.001), and their marital status (*p* = 0.003). Additionally, 68.8% (*n* = 989) of the participants believed (to a very high/high degree) the COVID-19 vaccine to be important for the health of their family members, friends, and communities, while 11.5% (*n* = 130) thought otherwise, which was significantly associated with age (*p* < 0.001) and employment (*p* = 0.045). In addition, 58.3% (*n* = 662) of the participants rated their level of knowledge about vaccination as very high or high, while only 13.9% (*n* = 158) of them rated their knowledge level as low or very low. Moreover, 72.7% (*n* = 829) of the participants encouraged their family members to get the COVID-19 vaccine, while only 9.5% (*n* = 107) of them strongly disagreed/disagreed with this issue; this was significantly associated with gender (*p* = 0.003) and age (*p* < 0.001). Interestingly, and as shown in Table 4, 49.4% (*n* = 561) of the participants declared that they would take the COVID-19 vaccine if it was not mandatory, while 34.8% (*n* = 395) would not take it; this was significantly associated with gender (*p* = 0.005), age (*p* =0.008), and employment (*p* = 0.035). Furthermore, 59.1% (*n* = 671) of participants intended to take the third dose, while 14.3% (*n* = 162) declared that they would not take it. Only 42.3% (*n* = 480) of the participants believed that the COVID-19 vaccination should be mandatory for the public, while 39.7% (*n* = 451) of them did not; this was significantly associated with gender (*p* < 0.001), age (*p* = 0.021), and employment status (*p* = 013). Importantly, only 23% (*n* = 261) of the participants had received vaccination against influenza in the past or in the current season, while 74.1% (*n* = 842) did not. Furthermore, 15.1% (*n* = 171) of the participants had refused to take certain vaccines in the past, while 80.3% (*n* = 912) of them had not. In addition, 47.9% (*n* = 544) of the participants preferred to wait until they had more information about these new COVID-19 vaccines, while 28.2% (*n* = 320) did not.

### 3.5. Assessment of the Side Effect of COVID-19 Vaccines

Table 5 shows the adverse effects of COVID-19 vaccination among the study participants. In this regard, 80.8% (*n* = 918) of the participants showed that they did not have any difficulties attributed to COVID-19 vaccine administration. Most of the participants had pain at the injection site (*n* = 808; 71.1%) with the first dose, second dose (752; 66.2%), and third dose (174;15.3%). Fever was also reported by 44.9% (*n* = 510), 36.8% (*n* = 418), and 9.0% (*n* = 102) of the participants after taking the first, second, and third doses of the vaccine, respectively. Headache was suffered by 22.8% (*n* = 259), 52.9% (*n* = 601), and 3.8% (*n* = 43) after the first, second, and third doses of the COVID-19 vaccine, respectively. Lethargy and fatigue were reported by 55.2% (*n* = 627), 39.6% (*n* = 450), and 10.7% (*n* = 122) after the first, second, and third doses of the vaccine, respectively (Table 5).

## 4. Discussion

This study aimed to provide data that would help in assessing the acceptance, motivators, and barriers for taking the COVID-19 vaccines, as well as assessing adverse effects of the vaccine. The main findings of the current study revealed that Pfizer was the most frequently administered vaccine. The acceptance of the COVID-19 vaccines was high, and people showed high confidence in its efficacy and safety. Most of our participants have positive attitudes and advocate in the fight against COVID-19. The study showed a high level of acceptance of COVID-19 vaccination among the study participants. Our study sheds light on the population’s intent to get vaccinated, as there are limited studies that have assessed COVID-19 vaccine acceptance, motivators, barriers, and side effects, both globally and in the western region of SA. Moreover, our study could help decision-makers locally, regionally, and internationally in planning future vaccine campaigns.

The current study was conducted on people living in the western region of SA. The rate of acceptance of COVID-19 vaccination for our participants was 53.2%. The studies that represented acceptance rates similar to our study were performed in Turkey, Ethiopia, and Malta (60.1%, 59.4%, and 50%, respectively) [15,16,17]. In addition, the studies that represented acceptance rates higher than our findings were performed in Canada, China, Malesia, Uganda, the US, Italy, Rome, and the Arabs region (95%, 93%, 83.3%, 70.1, 67%, 67%, 65%, and 62.4%, respectively) [14,18,19,20,21,22,23,24]. Moreover, the studies that represented acceptances rate lower than our findings were performed in southern Ethiopia, Turkey, and Egypt (64.1%, 34.6%, and 21%, respectively) [25,26,27].

Approximately 47% of our participants had seen or heard news and information about COVID-19 vaccination from social media, and 30% from local TV news. In this regard, the study conducted among the population in Ethiopia found that 33.7% of participants obtained their information about COVID-19 from mass media, 32.9% from the Internet, and 31.8% from social media [28], which was lower than our findings concerning social media.

The majority of our participants (74.6%) personally knew someone who had either contracted or died from COVID-19 infection. In this regard, a recent study found that 75.5% and 89.8% of medical and dental students personally knew someone who had contracted or died from COVID-19 infection, respectively [12], and the medical students’ data were similar to ours.

The confidence level among our participants concerning the COVID-19 vaccine efficacy was average (55.5%), and this is similar to the study which was conducted among university students in France regarding conventional vaccines (excluding COVID-19 vaccines; 52%) [13]. In addition, 68.8% of our participants very highly/highly considered that receiving the COVID-19 vaccine would protect the health of their family members, friends, and communities. A study performed in low and middle-income countries found that 88.1% of the participants considered that the COVID-19 vaccine would protect others [29], and this was slightly higher than our findings.

Another finding in our study was that 72.7% of participants agreed to encourage their family members to get the COVID-19 vaccine. A study performed in Ethiopia found that only 50.0% of participants would encourage their family/friends/relatives to receive the vaccine [28]. Another study among Lebanese dentists reported that 87.1% of their participants would have encouraged their family members to get the COVID-19 vaccine [30], which was higher than our findings.

About 42.3% of our participants supported and agreed on mandating COVID-19 vaccination of the public, while a study in Turkey [26] reported that only 27.1–29.5% of the participants thought that vaccination should be obligatory, which was lower than our findings. On the other hand, a study conducted on dental and medical students that found 40.3–67.9% of dental and medical students agreed/strongly agreed that COVID-19 vaccination should be mandatory for the public [12].

Only 23% of our participants had received the vaccination against influenza in the past or the current season. In this regard, a study performed in China found that only 14.6% of the participants had received vaccination against influenza in the past season [14], which was lower than our findings. Additionally, 15% of our participants had refused to take certain vaccines in the past, which was lower than the number found by a study performed in China (22.3%) [14]. Our findings showed that nearly half (47.9%) of the participants would have preferred to wait until they had more information about these new COVID-19 vaccines. In this regard, a previous study [13] showed that 56% of the participants wanted more information about the vaccine, which was close to our findings.

Almost 93.6% of our participants wanted to return to normal life as soon as possible, similarly to a report from France (85%) [13]. In addition, approximately 78.3% of participants in our study wanted to be actors in the fight against COVID-19, and 95.5% did not want to transmit COVID-19 to others, which was higher the findings of another report [13]. Furthermore, 63.4% of our participants wanted their children to resume school as soon as possible, which was much higher than a study performed in Lebanon (23%) [30]. In addition, 23.2% of our participants took the vaccine because it was mandatory for traveling abroad, which was lower than a study conducted in Lebanon (38%) [30]. Moreover, 24.2% of our participants did not delay getting a vaccine for reasons other than illness or allergy, which was akin to a Lebanese study (21.6–24.1%) [12].

A doctor’s recommendation was an important factor in vaccination decision-making in only 33.3% of our participants, while this figure was 80% in China [14], which is much higher than our findings. Additionally, 77.4% of our participants took the vaccine because it was recommended by health authorities (WHO and the Ministry of Health), which is much higher than a study conducted in Lebanon (45%) [30]. In addition, 58.9% of our participants were afraid of the adverse effects of the COVID-19 vaccine (e.g., fever, pain at the injection site, hospitalization), which was higher that a report from low and middle-income countries (41.2%) [29].

Regarding our study participants’ motivators and barriers regarding COVID-19 and its vaccine, we found that vaccine convenience (vaccination method, frequency, distance to vaccination sites, etc.) was an important factor in vaccination decision-making for 59.4% of them. In this regard, a study performed in China found that vaccine convenience was an important factor in vaccination decision-making among 75.7% of the participants (15), which was higher than our findings. In addition, 51.6% of our participants did not really understand how the COVID-19 vaccine works, while this number was only 10% in France [13].

Pain at the site of injection was the most common adverse effect that happened to our participants after taking the first, second, and third doses of the vaccine, at frequencies of 71.1%, 66.2%, and 15.3%, respectively. Furthermore, lethargy and fatigue were the second most common adverse effect after the first and third doses, at frequencies of 55.2% and 10.7%, respectively. On the other hand, headache was the second most common side effect after the second dose (52.9%). In addition, fever was considered the third most common side effect after taking the first and third doses of the vaccine, at frequencies of 44.9% and 10.7%, respectively. Moreover, lethargy and fatigue were the third most side effects after the second dose (39.8%). Finally, fever was reported after the second dose by 36.8% of participants.

### Strengths and Limitations of the Study

This study has several strengths, including the large number of participants. Additionally, there is a scarcity or a near lack of reports of such data from the population in the western region of SA. In addition, although our study is a common observational cross-sectional study, such studies represent the basis for preliminary information which is useful for policy implementation, as well as an indication of how well a policy would succeed. Moreover, measuring vaccine acceptance is essential for predicting vaccine campaigns success in view of the hesitancy associated with the novel COVID vaccine platforms. On the other hand, this is a cross-sectional study which is exploratory in nature, and it was conducted at a specific time point. Although this method has been widely used and accepted in published literature, the use of the online survey has some limitations. Participants may refuse to participate or have exaggerated or understated their self-reported vaccine-related adverse events. The online nature of the survey may also have limited the participation of older, illiterate individuals, or those who have no internet or social media access. This could lead to selection bias. Therefore, the data should be interpreted with caution. In addition, it is difficult to estimate the response rates among the studied population when using online surveys. Another limitation was that a larger percentage of the respondents were from a single geographic area, which may impact the generalization of the survey results. However, this would not affect the generalizability within the western region of Saudi Arabia. This is because the general demographic features of the population in the western region are consistent and homogeneous, since the same tribes and families living in the region are extended across the governorates of Makkah and Madeenah, as well as the biggest cities of Makkah, Jeddah, Taif, and Madeenah.

## 5. Conclusions

The current study showed a high level of acceptance of COVID-19 vaccination among people living in the western region of SA. Health education and communication from authoritative sources are important for alleviating public concerns about COVID-19 vaccine safety. The study was conducted only in the western region of KSA, and so the results may not represent the four districts of the kingdom. Future studies among residents of the four districts of Saudi Arabia are essential.

## Figures and Tables

**Table 1 vaccines-10-02097-t001:** Demographic data.

Questions	Category	Frequency *n* (%)
**Gender**	Male	576 (50.7)
	Female	560 (49.3)
**Age**	18–30	669 (58.9)
	31–40	198 (17.4)
	41–50	170 (15.0)
	51 or more	99 (8.7)
**Education level**	University	864 (76.1)
	High school	180 (15.8)
	Intermediate	35 (3.1)
	Master or Doctorate	57 (5.0)
**Residency city/area**	Taif	777 (68.4)
	Makkah	85 (7.5)
	Jeddah	123 (10.8)
	Yanbu	21 (1.8)
	Madinah	48 (4.2)
	Other	82 (7.2)
**Marital status**	Single	654 (57.6)
	Married	458 (40.3)
	Divorced	12 (1.1)
	Widow	12 (1.1)
**Employment**	Working in the private sector	112 (9.9)
	Working in the government sector	320 (28.2)
	Student or does not currently work	704 (62.0)
**Does anyone in the family work in the health sector?**	Yes	455 (40.1)
	No	671 (59.1)
**Where have you seen or heard news and information about COVID-19 vaccination? Please select all that apply**	Social media (e.g., Facebook, Twitter, YouTube)	1047 (47)
	Newspaper	180 (8.2)
	Magazine	91 (4.1)
	Radio	201 (9.1)
	Local TV news	670 (30)
**Do you have any of the following diseases?**	No comorbidities	1074 (94.5)
	Obesity	7 (0.6)
	Diabetes	2 (0.2)
	Hypertension	2 (0.2)
	Cancer	1 (0.1)
	Immunological disorders	3 (0.3)
	Respiratory disease (asthma)	4 (0.3)
	Other	43 (3.9)

**Table 2 vaccines-10-02097-t002:** Participants’ information about COVID-19 and its vaccine.

Questions	Category	Frequency *n* (%)
**Have you taken the COVID-19 vaccine?**	Yes	1054 (92.8)
	No	82 (7.2)
**What type of vaccine did you get in your doses?**	Pfizer	827 (72.8)
	AstraZeneca	256 (22.5)
	Moderna	37 (3.3)
	Others	2 (0.18)
**Reason of vaccine taking?**	I did not get it yet	14 (1.2)
	To protect myself and my family	835 (73.5)
	To receive services that are restricted to taking the vaccine	220 (19.3)
	To be with the community	114 (10.0)
	Others	14 (1.2)
**I personally know someone who had COVID-19 infection or died from COVID-19 infection**	Yes	847 (74.6)
	No	186 (16.4)
	I do not know	95 (8.4)

**Table 3 vaccines-10-02097-t003:** Participants’ motivators and barriers about COVID-19 and its vaccine.

Section	Response N (%)			*p*-Value						
	Yes	No	NA	Sex	Age	EL ^a^	RES	MS	EMP	WHS
I want to return to normal life as soon as possible.	1063 (93.6)	36 (3.2)	37 (3.3)	0.331	0.276	0.363	0.271	0.231	0.121	0.288
I want to be an actor in the fight against COVID-19.	889 (78.3)	94 (8.3)	153 (13.5)	0.246	0.052	0.761	0.007 **	0.007 **	0.376	0.873
I want my children to resume school as soon as possible.	720 (63.4)	100 (8.8)	316 (27.8)	0.352	<0.001 **	0.071	0.004 **	<0.001 **	<0.001 **	0.391
I took the vaccine to avoid infection with COVID-19.	820 (72.2)	184 (16.2)	132 (11.6)	0.299	0.025 *	0.143	0.011 *	0.262	0.243	0.346
I do not want to transmit COVID-19 to others.	1089 (95.9)	17 (1.5)	30 (2.6)	0.634	0.634	0.614	0.783	0.776	0.548	0.460
I took the vaccine because it is free of charge.	401 (35.3)	452 (39.8)	283 (24.9)	0.005 **	<0.001 **	0.019 *	0.200	<0.001 **	0.002 **	0.908
I took the vaccine because it is mandatory for traveling abroad.	263 (23.2)	576(50.7)	297 (26.2)	0.019 *	0.141	0.037 *	0.796	0.142	0.003 **	0.781
Social networks (e.g., Facebook, Twitter) have dissuaded me from getting vaccinated against COVID-19.	563 (49.6)	422 (37.1)	151 (13.3)	0.650	0.469	0.894	0.048 *	0.438	0.691	0.884
As an adult, have you ever delayed getting a vaccine for reasons other than illness or allergy?	275 (24.2)	760 (66.9)	101 (8.9)	0.786	0.547	0.943	0.081	0.491	0.789	0.661
For me, doctor’s recommendation was an important factor in vaccination decision-making.	378 (33.3)	516 (45.4)	242 (21.3)	0.091	0.008 **	0.004 **	0.222	0.098	0.331	0.548
I took the vaccine because it is recommended by health authorities (WHO, Ministry of Health).	879 (77.4)	180 (15.8)	77 (6.8)	0.603	0.125	0.052	0.046 *	0.979	0.545	0.432
I am afraid of side effects (e.g., fever, pain at the injection site, hospitalization) of the COVID-19 vaccine.	669 (58.9)	385 (33.9)	82 (7.2)	<0.001 **	0.876	0.761	0.902	0.902	0.240	0.738
Vaccine convenience (vaccination method, frequency, distance to vaccination sites, etc.) was an important factor in vaccination decision-making.	675 (59.4)	297 (26.1)	164 (14.4)	0.127	0.528	0.978	0.020 *	0.002 **	0.423	0.370
I do not really understand how the COVID-19 vaccine works.	586 (51.6)	418 (36.8)	132 (11.6)	0.863	0.011 *	0.003 **	0.432	0.019 *	0.197	0.018 *
I think the development of the COVID-19 vaccines seems to me to be too fast and that can be a barrier for me.	429 (37.8)	466 (41.0)	242 (21.2)	0.129	0.129	0.200	0.192	0.009	0.226	0.682

**^a^** EL: Education level, RES: residency area, MS: marital status, EMP: employment, WHS: anyone in the family work in the health sector, NA: not applicable. * Significant; ** highly significant.

**Table 4 vaccines-10-02097-t004:** Participants’ acceptance of COVID-19 and its vaccine.

Section	Response N (%)					*p*-Value						
	Very high	High	Normal	Low	Very low	Sex	Age	EL ^a^	RES	MS	EMP	WHS
What is your confidence in the efficacy of the COVID-19 vaccines?	241 (21.2)	385 (33.9)	343 (30.2)	79 (7.0)	88 (7.7)	<0.001 **	0.092	0.001 **	0.403	0.003 **	0.015	0.821
How important is it to you that getting the COVID-19 vaccine would protect the health of your family members, friends, and communities?	426 (37.5)	563 (31.3)	224 (19.7)	53 (4.7)	77 (6.8)	0.060	<0.001 **	0.045 *	0.264	0.075	0.182	0.250
How would you rate your level of knowledge about vaccination?	212 (18.7)	450 (39.6)	316 (27.8)	124(10.9)	34 (3.0)	0.096	0.046 *	0.006 **	0.381	0.096	0.107	0.534
	**SA**	**Agree**	**Normal**	**Disagree**	**SD**	**Sex**	**Age**	**EL**	**RES**	**MS**	**EMP**	**WHS**
I encourage my family members to get the COVID-19 vaccine.	482 (42.2)	347 (30.5)	200 (17.6)	45 (4.0)	62 (5.5)	0.003 *	<0.001 **	0.050	0.026	0.019	0.254	0.112
	**Yes**	**No**	**I don’t know**			**sex**	**Age**	**EL**	**RES**	**MS**	**EMP**	**WHS**
If COVID-19 vaccine was not mandatory, would you take it?	561 (49.4)	395 (34.8)	180 (15.8)			0.005 **	0.008 **	0.502	0.565	0.098	0.035 *	0.859
If the third dose became available, would you take it?	671 (59.1)	162 (14.3)	303 (26.7)			0.125	0.190	0.745	0.008 **	0.745	0.152	0.546
The COVID-19 vaccination should be mandatory for the public.	480 (42.3)	451 (39.7)	205 (18.0)			<0.001 **	0.021 *	0.013*	0.118	0.421	0.277	0.478
I received vaccination against influenza in a past season or in this season.	261 (23.0)	842 (74.1)	33 (2.9)			0.052	0.037 *	0.607	0.047 *	0.071	0.016	0.320
I refused to take certain vaccines in the past.	171 (15.1)	912 (80.3)	53 (4.7)			0.221	0.357	0.234	0.954	0.391	0.662	0.260
I prefer to wait until I have more information about these new COVID-19 vaccines.	544 (47.9)	320 (28.2)	272 (23.9)			0.021 *	0.543	0.039 *	0.010 *	0.250	0.210	0.342

**^a^** EL: Education level, RES: residency area, MS: marital status, EMP: employment, WHS: anyone in the family working in the health sector, SA: strongly agree, SD: strongly disagree. * Significant; ** highly significant.

**Table 5 vaccines-10-02097-t005:** Participants’ responses about the adverse effects of COVID-19 vaccines.

**Have you experienced any of these difficulties due to the COVID-19 vaccine? Please select all that apply.**	I did not have any problem	918 (8)
	Transportation issues	73 (6.4)
	Not being able to support the family	51 (4.5)
	Job loss	51 (4.5)
	Problems obtaining medications	24 (2.1)
	Problems with housing	25 (2.2)
	Others	34 (3.0)
**What were the side effects that happened to you after taking the first dose of the vaccine?**	Pain at the injection site	808 (71.1)
	Fever	510 (44.9)
	Headache	259 (22.8)
	Lethargy and fatigue	627 (55.2)
	Clotting	10 (0.9)
	Shortness and difficulty breathing	70 (6.2)
	Loss of consciousness and dizziness	46 (4.0)
	Others	12 (1.0)
	I did not suffer from any side effects	15 (1.1)
**What were the side effects that happened to you after taking the second dose of the vaccine?**	Pain at the injection site	752 (66.2)
	Fever	418 (36.8)
	Headache	601 (52.9)
	Lethargy and fatigue	450 (39.6)
	Clotting	7 (0.6)
	Shortness and difficulty breathing	86 (7.6)
	Loss of consciousness and dizziness	59 (5.2)
	Others	31 (2.7)
	I did not take the vaccine yet	26 (2.3)
**What were the side effects that happened to you after taking the third dose of the vaccine?**	Pain at the injection site	174 (15.3)
	Fever	102 (9.0)
	Headache	43 (3.8)
	Lethargy and fatigue	122 (10.7)
	Clotting	3 (0.3)
	Shortness and difficulty breathing	9 (0.8)
	Loss of consciousness and dizziness	5 (0.4)
	Others	8 (0.7)
	I did not take the third dose yet	918 (80.8)

## Data Availability

All data associated with this study are included herein.

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
