# Peer review of "Motivators, Barriers, and Acceptance of COVID-19 Vaccination among Residents of Western Saudi Arabia"

_vaccines, 2022, doi:10.3390/vaccines10122097_

Round 1

Reviewer 1 Report

The authors through a questionnaire study tried to find out the acceptance rate of covid-19 vaccination in a region of SA. The study question is clear and the answers to the questionnaire are straightforward. The acceptance rate was found to be satisfactory in the specific region studied and the side effects of vaccination were relatively low. Although the study is clear and the results are coming forward from the answers to the questionnaire, the present reviewer believes that it is only a statistical analysis for vaccination acceptance in one region of SA and nothing more than that. It would be interesting to know the type of vaccines that were used. Besides the results of the side effects were anticipated since it is known that covid-19 vaccines did not have any side effects. The present reviewer can not see any novelty or even something that promotes/or stops vaccination. The "opinion" about covid-19 vaccination can not be a study question.

Author Response

REVIEWER 1

Response to the General comment: We thank the reviewer for the time and effort evaluating the manuscript as well as commending the manuscript.  The vaccine used in Saudi Arabia included Pfizer-BionTech, Astra-Zenca and Moderna as well as others. Their frequencies are indicated in Table 2 of the manuscript.  Similar findings were shown in our recently accepted article (Almalki et al., Frontiers in Pharmacology, 2022). However, we disagree with the reviewer about the opinion on the novelty of the study. There are no reports of COVID-19 vaccine acceptance in the western region of SA, where the study was conducted. Also, COVID-19 vaccine acceptance studies are very popular in the literature in disagreement with the reviewer’s comment about the study question. Several of these studies are cited in the manuscript. In addition, although the vaccine adverse reactions are expected to be low, there is a scarcity or almost no reports of such data from the population in the western region of SA. This study presents a proof of such expectation.

Reviewer 2 Report

Methods

“The current study was a cross-sectional survey conducted among people living in the western region of SA who were conveniently invited to participate in this study. The study was conducted in the period from December 2021 to March 2022. Participants were sent a link for the study survey, and participation was voluntary.”

[please state your sampling frame (i.e. your eligibles and how you identified them) how many did not reply and how did the characteristics of those not replying compare to those who responded? This is key for your study]

“The current study tool was a self-administered questionnaire designed after consulting previously published studies. The questionnaire included five sections”

[please provided details of originals and questions in previous questionnaires. Any reliability and validity data?]

Results

“As shown, more than half of the participants (n=669; 58.9%) had an age range from 18-30 years (mean age: 34.5 SD: 9.8). Also, less than half of the participants were married (n = 458; 40.3%) and most of them had a college degree (n = 864; 76.1%). In this regard, participants were females (n = 560; 49.3%).” Also 68% from Taif city

[please emphasise in detail in the abstract, results and discussion sections that you can only generalise to the participants with these characteristics].

[Consider whether some of the tables are not of great interest to the reader and place those in a Supplemental file and comment in the text in a few sentences].

Author Response

REVIEWER 2

Comment #1: Methods: “The current study was a cross-sectional survey conducted among people living in the western region of SA who were conveniently invited to participate in this study. The study was conducted in the period from December 2021 to March 2022. Participants were sent a link for the study survey, and participation was voluntary.

” [please state your sampling frame (i.e. your eligible and how you identified them) how many did not reply and how did the characteristics of those not replying compare to those who responded? This is key for your study].

“The current study tool was a self-administered questionnaire designed after consulting previously published studies. The questionnaire included five sections” [please provided details of originals and questions in previous questionnaires. Any reliability and validity data?]

Response to Comment #1: We thank the reviewer for the time and effort reviewing the manuscript as well as commending the manuscript. As mentioned in the methods section, the participants were those who received the link for the questionnaire in the indicated period and voluntarily responded. Adults above the age of 18 who agreed to take part in the study were included in it. Participants under the age of 18 were not permitted. We do not know how many did see the link and did not respond or even identify those who responded. The survey did not collect in participant’s identifier. As suggested by the reviewer, references to previous questionnaires including validity of the survey questions were added to the revised manuscript. On the other hand, convenient samples are taken when there is no sampling frame. It is considered nonprobability sampling technique

Comment #2: Results: “As shown, more than half of the participants (n=669; 58.9%) had an age range from 18-30 years (mean age: 34.5 SD: 9.8). Also, less than half of the participants were married (n = 458; 40.3%) and most of them had a college degree (n = 864; 76.1%). In this regard, participants were females (n = 560; 49.3%).” Also 68% from Taif city [please emphasise in detail in the abstract, results and discussion sections that you can only generalise to the participants with these characteristics]. [Consider whether some of the tables are not of great interest to the reader and place those in a Supplemental file and comment in the text in a few sentences]. 

Response to Comment #2: We thank the reviewer for the comment, and we revised the manuscript to reflect her/his opinion about the discussion section. The above data are already included in the abstract. Also, we have revised the limitation section of the discussion. However, we see that all the five tables are essential to the study specially because the journal is an open access with no difference between supplemental data and those included in the original paper. The reader can skip the table (s)he is not interested.   

Reviewer 3 Report

This paper is based on the cross-sectional survey on the acceptance and side effects of vaccines related to COVID-19 in a relatively homogeneous sector of Saudi Arabia.  The description of the results is limited.  The paper has to improve the following sections:

1. Presenting a Theoretical Framework:  The study results could be framed from a selected theoretical framework such as the Knowledge-Motivation-Attitude-Practice model.  

2. Highlighting Important Contributions or Uniqueness of the Study: What is new from this study?  How can future research be redesigned to improve the relevance of behavioral factors influencing public health practice in disease prevention and treatment?

3. Clarifying the Measurement Scales:  It is unclear how the behavioral variables or scales are constructed.  The validity and reliability of the study scales should be carefully documented in the Methods Section.

4. Presenting the Generalizable Results:  If the results are not generalizable, it is important to document the limitations.

Overall, this is a limited study.  The lack of clarity in assessment methods and a theoretical framework to guide the analysis may have dampened my enthusiasm.

Author Response

 REVIEWER 3 General comment: This paper is based on the cross-sectional survey on the acceptance and side effects of vaccines related to COVID-19 in a relatively homogeneous sector of Saudi Arabia.  The description of the results is limited.  The paper has to improve the following sections:

Response to the General comment: We thank the reviewer for the time and effort evaluating the manuscript. We hope that the modification in the indicated sections and the data presentation in the revised manuscript meet the reviewer’s expectations.

Comment #1: Presenting a Theoretical Framework:  The study results could be framed from a selected theoretical framework such as the Knowledge-Motivation-Attitude-Practice model.  

Response to Comment #1: The results is currently classified into 5 sections as follows:

  1. Section 3.1: Demographic characteristics of the study participants
  2. Section 3.2: Participants' information about COVID-19 and its vaccine (knowledge)
  3. Section 3.3. Participants' acceptance of COVID-19 and its vaccine (Attitude and Practice)
  4. Section 3.4. Participants’ motivators and barriers for administration of COVID-19 vaccine (Motivation)
  5. Section 3.5. Assessment of the side effect of COVID-19 vaccines

Based on the reviewer’s suggestion, we substituted section 3 and 4 by each other.

Comment #2: Highlighting Important Contributions or Uniqueness of the Study: What is new from this study?  How can future research be redesigned to improve the relevance of behavioral factors influencing public health practice in disease prevention and treatment?

Response to Comment #2: The importance of the study was emphasized in the revised discussion section. Also, future perspectives were added to the discussion section.

Comment #3: Clarifying the Measurement Scales:  It is unclear how the behavioral variables or scales are constructed.  The validity and reliability of the study scales should be carefully documented in the Methods Section.

Response to Comment #3: The Methods section was revised to reflect the reviewer’s point of view.

Comment #4: Presenting the Generalizable Results:  If the results are not generalizable, it is important to document the limitations. Overall, this is a limited study.  The lack of clarity in assessment methods and a theoretical framework to guide the analysis may have dampened my enthusiasm.

Response to Comment #4: As indicated in the response to reviewer #2, we have revised the limitation section of the study. Also, the methods section was revised to improve the assessment methods.

Round 2

Reviewer 1 Report

Statistical analysis and observational studies are welcome, but the present reviewer believes that there is no need for publication the observational-statistical analysis of a specific region regarding vaccination. Acceptance or rejection rate of vaccination is upon legislational, regional rules in every Country

Author Response

REVIEWER 1

General comment: Statistical analysis and observational studies are welcome, but the present reviewer believes that there is no need for publication the observational-statistical analysis of a specific region regarding vaccination. Acceptance or rejection rate of vaccination is upon legislational, regional rules in every Country.

Response to the General comment: We have a point of view to the reviewer about the opinion on the publication of the study. As mentioned in the previous round, there are no reports of COVID-19 vaccine acceptance in the western region of SA, where the study was conducted. Also, COVID-19 vaccine acceptance studies are very popular in the literature in disagreement with the reviewer’s comment and several of these studies are cited in the manuscript. In addition, there is a scarcity or almost no reports of such data from the population in the western region of SA. Moreover, one of the advantages of this study is the large sample size. Importantly, although our study is a common observational cross-sectional study, such studies represent the bases for preliminary information useful for policy implementation and represent an indication of how far a policy would succeed or not. This last sentence was added to the Discussion section.

Reviewer 2 Report

Thanks to the authors for their appropriate revisions.

Author Response

REVIEWER 2

General comment: Thanks to the authors for their appropriate revisions.

Response to the General comment:  We thank the reviewer for accepting the revised manuscript.

Reviewer 3 Report

The amended manuscript is much improved.  However, the methods section could document the reliability and validity of the measurement scales used in this study. Without any documentation on the psychometric properties of the measurement scales, it will limit its scientific merit.

Author Response

REVIEWER 3

General comment: The amended manuscript is much improved.  However, the methods section could document the reliability and validity of the measurement scales used in this study. Without any documentation on the psychometric properties of the measurement scales, it will limit its scientific merit.

Response to the General comment:

This study was not trying to develop a universal tool for assessment of vaccine acceptance.  It is a normal straightforward descriptive study that aimed to describe the pattern of observations regarding vaccine acceptance in the western region of Saudi Arabia. It will be useful to inform policy makers and represent a lesson that can be used in similar countries. Psychometric measures are used when we are developing a scale or a tool that is constructed from a group of items and that can be used to assess behavioral aspects and categorizing/ranking respondents according to the levels  across behaviors such as Fagerstrom questionnaire for Nicotine dependence and Morisky scale for assessment of medication adherence and tools developed for assessment of quality of life. In this case, there are complex steps such as assessing construct validity and factor analysis. In our case it was enough to validate the questionnaire for face and contents to ensure that its easily understood and there is no ambiguity and that it is content wise covering all possible responses respondents should choose from. Our participants responses have been treated individually as separate variables as it is the normal practice in descriptive observational studies.